# Impact of Preemptive Postoperative Pressure Support Ventilation and Physiotherapy on Postoperative Pulmonary Complications after Major Cervicofacial Cancer Surgery: A before and after Study

**DOI:** 10.3390/medicina59040722

**Published:** 2023-04-06

**Authors:** Guillaume Salama, Cyrus Motamed, Jamie Elmawieh, Stéphanie Suria

**Affiliations:** Department of Anesthesia, Institut de Cancérologie Gustave Roussy, 94080 Villejuif, France

**Keywords:** cervicofacial surgery, oncologic surgery, respiratory complications, pressure support ventilation

## Abstract

*Introduction*: Complex cervicofacial cancer surgery with free flap reconstruction is known to have a high incidence of postoperative pulmonary complications (PPCs). We hypothesized that by implementing an optimized respiratory protocol, including preemptive postoperative pressure support ventilation, physiotherapy, and critical respiratory support and follow-up, we could decrease the incidence of PPCs. *Patients and methods*: We evaluated the incidence of PPCs over two periods in two groups of patients having a routine or optimized postoperative respiratory protocol: 156 adult patients undergoing major cervicofacial cancer surgery were assessed; 91 were in Group 1 (routine) and 65 were in Group 2 (optimized). In Group 1, no ventilatory support sessions were performed. The incidence of pulmonary complications in both groups was compared using a multivariate analysis. Mortality was also compared until one year postoperatively. *Results*: In Group 2 with an optimized protocol, the mean number of ventilatory support sessions was 3.7 ± 1 (minimum 2, maximum 6). The incidence of respiratory complications, which was 34% in Group 1 (routine), was reduced by 59% OR = 0.41 (0.16; 0.95), *p* = 0.043) to 21% for the optimized Group 2. No difference in mortality was found. *Conclusions*: The present retrospective study showed that using an optimized preemptive respiratory pressure support ventilation combined with physiotherapy after a major cervicofacial surgery could possibly help reduce the incidence of pulmonary complications. Prospective studies are needed to verify these findings.

## 1. Introduction 

Postoperative pulmonary complications (PPCs) are defined as events affecting the respiratory tract that can adversely alter the clinical outcome. Atelectasis is reported to be a major component of PPCs [1]. In major cervicofacial cancer surgery with reconstruction and free flap, patients necessitating postoperative tracheostomies are reported to have an incidence of PPCs as high as 32% [2]. On the other hand, pressure-assisted ventilation (PAV) might play a positive role under certain circumstances in laparoscopic surgery in comparison to spontaneous ventilation [3]. To our knowledge, PAV has not been assessed in major cervicofacial surgery as most of the cited research has focused on intraoperative ventilation and sometimes before extubation, while the postoperative period is seldom considered. We hypothesized that an optimized postoperative respiratory protocol including preemptive, short, and intermittent pressure-assisted ventilatory support could help decrease the incidence of PPCs in the postoperative period.

The primary objective of this study was to assess the incidence and factors influencing early PPCs after the implementation of a pressure support-based postoperative ventilatory protocol in patients after major cervicofacial cancer surgery with reconstruction who had a tracheostomy. The secondary objective assessed the one-year prognosis of patients after the implementation of the protocol.

## 2. Methods

This study is a monocentric, retrospective before and after observational study as part of a quality assurance program carried out at the Gustave Roussy Cancer Institute from 2 January 2019 to 1 November 2020. Our institutional review board approved and permitted publishing the results on 6 October 2020. All patients were informed by a letter that their anonymous data could be used in a scientific publication.

Inclusion criteria: All patients scheduled for major cervicofacial cancer surgery lasting more than 4 h and having a tracheostomy with a cuff cannula placed by surgeons with or without flap reconstruction.

Group 1 (2 January 2019 through 28 October 2019) was the historical control group with routine respiratory care, and Group 2 (3 November 2019 to 2 November 2020) was the study or optimized respiratory group having the same type of surgery.

All patients stayed in post anesthetic care unit (PACU), followed by surgical care unit (SICU), according to our routine local clinical protocol.

Intraoperative monitoring:

Intraoperative monitoring consisted of continuous electrocardiogram, non-invasive blood pressure monitoring, oxygen saturation, objective neuromuscular monitoring, bispectral index, and intraoperative central temperature monitoring, while respiratory parameters were recorded from the anesthesia machine, including ventilatory pressures, ventilatory volume, and end-tidal PCO_2_ for all patients. Invasive arterial lines were inserted after induction of general anesthesia if necessary according to our local protocol in cases of flee flap surgery, ASA III patients, long or complex surgeries, and patients with major cardiorespiratory and renal histories. All data were stored manually or automatically in our anesthesia information management system, Centricity Anesthesia^®^.

Respiratory management:

Group 1 had routine intra- and postoperative respiratory management at the discretion of anesthesia and intensive care providers, and no specific protocol was followed intraoperatively. Intraoperative ventilation was adjusted to maintain ETCO_2_ of 30–40 mmHg at the discretion of the anesthesia providers. Recruitment maneuvers were performed on demand if desaturation (<95%) occurred. In the postoperative period, all patients had spontaneous ventilation through tracheostomy cannula, with or without oxygen, to maintain an oxygen saturation above 95%, and physiotherapy was on demand at the discretion of the anesthesiologist/intensivist in charge.

As part of quality assurance program, Group 2 benefited from non-invasive preinduction ventilatory protocol that consisted of preoperative oxygenation with assisted pressure ventilation (2 cm H_2_O) to obtain a tidal volume (VT) of 6–8 mL/kg with 100% initial FiO_2_. During surgery, a protective ventilation protocol with a VT of 8 mL/kg and ventilatory frequency adapted to maintain an end-tidal CO_2_ of 25–40 mmHg was used. Recruitment maneuvers were performed every 30–45 min. Blood gas was drawn regularly to adjust ventilatory set up. The following postoperative respiratory optimization protocol, which consisted of sessions of preemptive pressure support ventilation 30 min upon arrival at PACU and was followed every 6 h for 24 h in SICU, was initiated. Oxygen titration proceeded to achieve a minimum of 95% saturation. Inspiratory pressure support was titrated to obtain a tidal volume between 6 and 8 mL/kg, and the cuff of the tracheostomy cannula was inflated during the ventilatory assistance and deflated after the end of the sequences. A physician set up the first episode of pressure support ventilation while intensive care nurses adjusted the following episodes. At least one session of physiotherapy in the SICU on day 1 was mandatory. Patients’ adhesion to protocol (number and tolerance) was recorded.

A standard heat humidifier was always operational for all patients, with the temperature set at 37 °C. The initial cuffed cannula was changed systematically at 48th postoperative hour by the cervicofacial surgeon and replaced with a non-cuffed cannula, which was finally removed or replaced again between 5th and 6th days in all patients depending on type of surgery and clinical indication.

The primary objective (PPC incidence) was defined as postoperative respiratory complications during the first five postoperative days, including pulmonary infections (hospital-acquired pneumonia) as defined by criteria of French Experts’ Society of Anesthesia and Resuscitation [4]. These criteria included radiological features, lung infiltrates, and at least one of the following items: temperature >38.3 °C unrelated to other causes, 4000 mm < blood white cells, and <12,000 mm. In addition, at least two of the following items were added: purulent sputum, cough or dyspnea, desaturation, declining oxygenation or increased oxygen requirement, hypoxemic atelectasis, and transfer to main intensive care unit (ICU) for any respiratory cause. A chest X-ray was performed to check the positioning of the nasogastric tube in the postoperative care unit (PACU) or at day 1 for all patients. If respiratory events were suspected, a chest X-ray was performed in conjunction with blood tests including C reactive protein, electrolytes, white blood cells, platelet counts, hemoglobin level, and glomerular filtration rate, which were performed daily during the stay in surgical intensive care unit (SICU); blood gas was performed if requested, depending on clinical situations.

### 2.1. Non-Respiratory Care and Data Extraction

All patients transited from PACU to SICU. For postoperative management, all patients stayed in SICU for at least one night, those having free vascular flap had a minimum stay of 3 nights, which focused on optimizing analgesia, hemodynamics, respiratory surveillance, and quick screening of medical and surgical complications. Clinical data extraction was obtained from three different databases available in our hospital, Centricity Anesthesia GE^®^, Ambre^®^, and DX Care^®^. The following parameters were collected: age, gender, body mass index (BMI), and medical history. Assessment of malnutrition was defined by weight loss ≥5% in 1 month or ≥10% in 6 months, body mass index (BMI) < 21 kg/m^2^ or albuminemia <35 g/L, and addiction assessment: active or weaned smoking status and/or alcoholism. Preoperative morphine, location of the tumor, and history of radiotherapy and/or chemotherapy and/or preoperative immunotherapy. Preoperative data included medical history and systemic medications and opioids, if present. Intraoperative data collected include the type of surgery, duration of surgery, regional anesthesia for postoperative pain (if performed), hemodynamic parameters, administration of fluids and blood products, and ventilation parameters (FiO_2_, mean tidal volume, and mean end-expiratory pressure (PEEP) after induction). Postoperative parameters include survival rates at postoperative days 1 and 30, 6 months, and 1 year, as well as length of stay in the operating room and ICU. Postoperative parameters included Clavien–Dindo classification of surgical complications [5] (Table 1), and other non-respiratory medical complications, which were mainly infections, neurologic, metabolic, cardiovascular, gastrointestinal, renal failure, and electrolytic disturbances.

### 2.2. Statistical Analysis

Patients’ characteristics are reported using numbers and percentages for qualitative variables (including binary variables) and were compared by Chi-squared test, while means with standard deviation for quantitative variables were compared using Student t-test. Standard mean differences (SMDs) were used to compare variables distributions between groups; an SMD greater than 0.1 and suggesting an uneven distribution was followed by an adjustment during multivariate analysis.

Main objective: Up to six multivariate logistic models were applied to explain PPCs. We adjusted all variables having SMD > 0.1 (model A), SMD > 0.15 (model C), and SMD > 0.2 (model E). The stepwise regression technique (using step AIC from the MASS package in R) was then used to get more parsimonious models (model B, model D, and model F). The best model was then chosen among these 6 models using the best Bayesian information criterion. The effect of PAV was finally compared in all 6 models to ensure that model selection does not change the results (sensitivity analysis). For the secondary objective, comparisons were made using statistical tests without adjusting to alpha error. A univariate Cox model and graphic of cumulative incidence were constructed in both groups in order to compare mortality in POD 30, 6 months, and 1 year.

Sample size was chosen in order to obtain similar incidence of PPCs using previous studies with comparative types of surgeries in a limited period [2,6].

## 3. Results

One hundred and fifty-six patients were finally included in the study. Figure 1 displays the flowchart of the patients: Group 1 consisted of 91 patients, and Group 2 consisted of 65 patients. All the patients in Group 2 participated and tolerated the protocol well. Group 1 did not have any pressure support ventilation during their stay in the SICU.

All Group 2 patients had a mean 3.7 ± 1 sessions of pressure support ventilation (minimum two and maximum six) in the first postoperative day, and none of them presented clinical signs of overventilation such as hypotension, confusion, or weakness. All patients in this group also had at least one session of physiotherapy on postoperative day 1.

Patient characteristics: The mean age of patients was 59.5 ± 9 years (minimum 24 and maximum 91), with a prevalence of 66 and 60% for the male gender, see Table 2. A practice change for triggering redpack cell transfusion from 10 to 9 g/dL as part of another quality assurance program was noticed at the end of 2019, as was a surge in the practice of regional anesthesia for harvest sites of free flap whenever it was indicated. Tobacco use was similar in both groups; however, active alcoholic consumption was more important in Group 1, see Table 2. Cancer of the oral cavity was present in 73.6% of cases in Group 1 and in 87.7% in Group 2.

Intraoperative events are displayed in Table 3.

Postoperative complications:

The incidence of postoperative respiratory complications was 37% in Group 1 (routine) and 21.5 in Group 2 (optimized), which was significantly different between groups *p* = 0.034, Table 4.

No tracheal cannula was plugged during the stay in SICU.

Postoperative pressure-assisted ventilation was associated with a decreased risk of the occurrence of respiratory complications of 59% (OR = 0.41 [0.16; 0.95]) in a multivariate analysis. Using multiple selected variables, six statistical models were obtained (Table 5); in all models, non-invasive ventilation had a protective effect on PPCs, with an OR < 1. The duration of surgery had a negative impact (Figure 2), and the excessive positive fluid balance was deleterious in some models. In order to predict PPCs, a stepwise regression was performed. Thereafter, for checking the robustness of the results, the logistic model construction was repeated, taking into account other significant thresholds between the groups’ standard mean difference (SMD) > 0.15, then SMD > 0.2 in a stepwise regression procedure, and in the final model, F was selected.

The incidence of other postoperative medical complications and secondary transfer to the ICU was 39.5 and 25.7% in Group 1 vs. 27.7 and 24.4% in Group 2, which are not significantly different (*p* = 0.124). One death occurred during the overall hospital stay in each group. The mean length of the overall hospital and SICU stays were not different: 25.7 ± 9.3 and 4.8 ± 3 days in the control group vs. 24.4 ± 6.8 and 4 ± 1.5 days in the optimized group (*p* = 0.33 and *p* = 0.05).

Surgical complications were not significantly different between the groups, with 51.6% in Group 1 versus 64.6% in Group 2 (Table 6).

Mortality:

Thirty days of hospital mortality were similar, with one person dead in each group.

Mortality after 1 year in Group 2 was 13.5% versus 38.9% in Group 1; however, since the duration of the follow up was not similar, a Cox cumulative analysis was performed and no statistically significant difference was noticed, *p* = 0.078, Figure 3.

## 4. Discussion

This study suggests that postoperative preemptive pressure support ventilation and physiotherapy in major cervicofacial cancer patients with tracheostomies were independent protective factors of the postoperative respiratory event, instigating a significant reduction in the incidence of PPCs. The duration of surgery was found to be the only independent factor that was associated with an increased risk of pulmonary complications, especially when surgery lasted more than 10 h. The incidence of PPCs in our patients was in accordance with a previous investigation that reported results between 15% and 45% [7]. Despite reports suggesting that pressure support ventilation facilitates weaning from mechanical ventilation in the intensive care unit, few studies have assessed its effects on recovery from anesthesia. In a randomized study, Jeong et al. [3] focused on emergence from anesthesia and showed that when compared to spontaneous emergence, the pressure support ventilation reduced postoperative atelectasis in patients undergoing a laparoscopic colectomy or robot-assisted prostatectomy. In their study, these authors applied this mode of ventilation in the PACU, whereas in our study, we also applied it intermittently in the SICU for 24 h. Although the major limitation of this latter study [3] was that the postoperative atelectasis diagnosis was performed with a sonography, which requires personal skill for an accurate assessment, but also a patient’s compliance. In a recent prospective study, there was no correlation between PPCs and age, smoking history, comorbidities, type of resection, of for major cervicofacial surgery patients [7]. The lung protection ventilatory protocol for obese patients in general is reported to be more efficient with volume-controlled ventilation and individualized positive-end expiratory pressure [8]. Additionally, recently it has been reported that total intravenous anesthesia with propofol might also have a protective pulmonary effect on cervicofacial cancer surgery [9], but some methodological issues in this recent study necessitate further investigations. Protective lung ventilation was also part of our protocol; however, there was no significant difference in applying a low-tidal volume in both groups, as protective low-tidal volume ventilation is now more and more commonly used by anesthesiologists; therefore, we assume that the main difference in ventilatory management between the two groups was the preemptive pressure-assisted ventilation that was applied in the first few postoperative hours. To our knowledge, there is no report describing the effects of preemptive pressure support ventilation in major cervicofacial surgery with tracheostomy in the postoperative period.

We believe that it was rational to use pressure support ventilation in this category of patients with a relatively high incidence of PPCs, as this complication has a multifactorial origin and therefore, prevention might yield a better outcome than treatment [10,11]. On the other hand, the beneficial impact of postoperative physiotherapy is mostly described in abdominal surgery [12], and data are sparse in major cervicofacial surgery. One can also question the real impact of a few 30 min sessions of preventive support ventilation and clinically relevant efficiency in this context. We believe that these sessions probably treated infraclinic atelectasis and probably contributed to the positive outcomes. The use of continuous airway positive pressure (CPAP), which is in fact a less invasive procedure could also be discussed in this context [13]; however, the higher risk of barotraumatism, especially in this category of patients with a high percentage of COPD history, should also be balanced. Accordingly, due to the positive results of this quality assurance program, the study inclusion protocol is now the protocol for our PACU/SICU admission. Nevertheless, we are aware that some complex surgeries with a shorter duration or cervicofacial interventions in relation to other types of cancer may need other types of assistance [14], and our protocol is mainly adapted to our specific category of patients needing mostly free flap reconstruction and postoperative tracheostomies. Indeed, the involvement of the lower or upper respiratory tract can totally change reconstruction modalities [15]; therefore, transitory tracheostomy may not be necessary, allowing facemask non-invasive ventilatory assistance or even CPAP in selected patients.

Postoperative tracheostomy could also play an additional role in the occurrence of PPCs in relation to a salivary stasis favoring micro inhalations [16]. The necessity to inflate the cuff during pressure-assisted ventilation could provide an explanation for the fewer incidences of PPCs in the optimized Group 2; however, an average maximum of 2 h of cuff inflation in the first 24 h could hardly be a significant determinant of the decrease in PPCs. The incidence of abusive alcohol consumption in Group 1 (routine) was significantly more important compared to Group 2 (optimized). This difference could have produced a selection bias, since alcohol exposure may have contributed to an increase in PPCs by promoting an altered immune system and increased inflammatory responses and PPCs [17,18]. On the other hand, the percentage of patients who had a preliminary oncologic treatment before surgery were significantly more important in Group 2 (33.8%) versus 22% in Group 1. This could have worsened the incidence of pulmonary infections, and hemodynamic instability as radiotherapy, for example, can disturb the baroreflex or prolong surgery [19,20,21]. With regard to other non-respiratory medical complications, their respective incidences of 39.5% and 27.7% are in accordance with the literature [22,23,24]. This protocol did not change the one-year mortality rate, which was probably because of the lack of power of the study.

This study is one of the very few that assesses the effect of a systematic respiratory optimization protocol using PAV on PPCs in major cervicofacial surgery cancer patients. In addition, the enhanced statistical methodology permitted the display of robust results. Indeed, the primary objective was the same in all models. Nevertheless, some shortcomings must be listed. Indeed, a changing practice such as a decreasing critical hemoglobin level before transfusion and the increased practice of regional anesthesia for postoperative analgesia explaining the lesser use of opioid in the optimized patients could be considered confounding factors. In addition, silent atelectasis without hypoxemia was not included because, beside the first postoperative day, no chest X-ray was proposed if no clinical signs were found. Moreover, the duration of surgeries was somehow longer in comparison with other studies [25,26,27] since our institution is a teaching super-specialized hospital. We also focused only on PPCs until the fifth postoperative day; however, these patients have multiple symptom burdens, which may appear later in the hospitalization course [28] and which probably overlap with each other, and so focusing on only one event in a limited time frame might not be sufficient to significantly change the prognosis and outcome. Finally, the retrospective collection of data and the monocentric specificity of this study do not permit a generalization of the results.

## 5. Conclusions

In this study, respiratory optimization consisting of protective intraoperative lung ventilation, preemptive postoperative pressure support ventilation, and physiotherapy in major cervicofacial cancer surgery with a tracheostomy were associated with a significant decrease in PPCs in our group of patients. In addition, the duration of surgery was a confounding factor for this complication. Future multicentric prospective studies are necessary to confirm our findings.

## Figures and Tables

**Figure 1 medicina-59-00722-f001:**
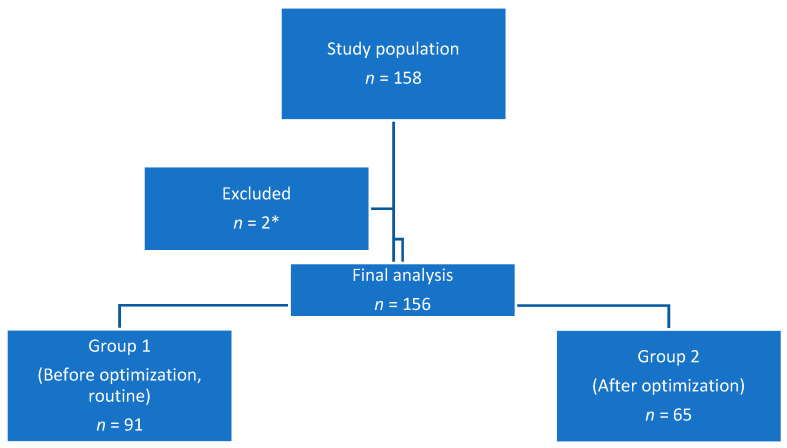
Study flowchart. * Exclusions concerned 2 patients who were below 18 years old at the time of the study.

**Figure 2 medicina-59-00722-f002:**
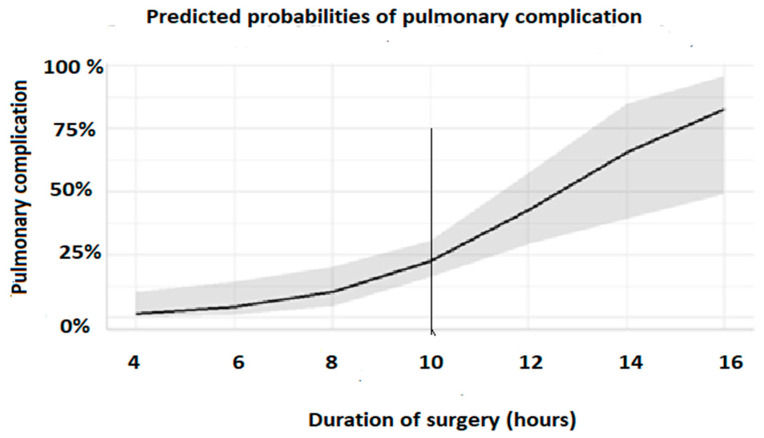
Impact of duration of surgery on pulmonary complications. After 10 h of surgery (cut off line) the probability of pulmonary complication significantly increased.

**Figure 3 medicina-59-00722-f003:**
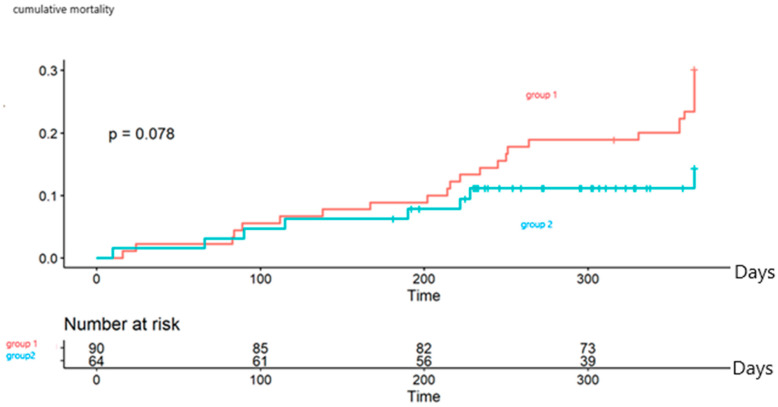
Cox cumulative analysis of mortality at 1 year. No statistical significance was noticed.

**Table 1 medicina-59-00722-t001:** Clavien–Dindo classification of surgical complications.

Grade	Definition
Grade I	Any deviation from the normal postoperative course without the need for pharmacological treatment or surgical, endoscopic, and radiological interventions. The allowed therapeutic regimens are as follows: drugs as antiemetics, antipyretics, analgetics, diuretics, electrolytes, and physiotherapy. This grade also includes wound infections opened at the bedside.
Grade II	Requiring pharmacological treatment with drugs other than such allowed for Grade I complications. Blood transfusions and total parenteral nutrition are also included.
Grade III	Requiring surgical, endoscopic, or radiological intervention.
–IIIa	Intervention not under general anesthesia.
–IIIb	Intervention under general anesthesia.
Grade IV	Life-threatening complication (including Central Nervous System complications) requiring IC/ICU-management.
–IVa	Single-organ dysfunction (including dialysis).
–IVb	Multiorgan dysfunction.
Grade V	Death of a patient.

**Table 2 medicina-59-00722-t002:** Demographic and preoperative characteristics.

	Group 1 (*n* = 91) Routine	Group 2 (*n* = 65) Optimized	*p* Value	SMD
Sex male/female, *n* (%)	60/31 (66/34)	39/26 (60/40)	0.54	0.075
Age (mean ± SD)	59 ± 10	60 ± 8	0.5	0.151
BMI (mean ± SD)	28.7 ±12	24.1 ± 4	0.001	0.124
Malnutrition, *n* (%)	20 (22)	10 (15.4)	0.37	0.169
Tobacco history, yes, *n* (%)	56 (61.5)	37 (57)	00.2	0.122
Weaned	35	27	0.64	0.035
Metabolic diseases, yes, *n* (%)				0.155
Non-weaned alcoholism	28 (30.8)	10 (15.4)	0.01	
Diabetes	12 (13.2)	9 (13.8)	0.8	
Pulmonary disease, yes, *n* (%)				
COPD (*n*)	6 (6.6)	5 (7.7)	0.83	0.175
Respiratory insufficiency, *n* (%)	0	1 (1.5)		
Asthma	4 (4.4)	1 (1.5)		
Lung cancer	0	2 (3)		
SAS	2 (2.2)	2 (3)		
Others	2 (2.2)	1 (1.5)		
Cardiovascular disease, yes, *n* (%)				
Hypertension	30 (32.3)	27 (41.5)	0.2	0.169
Myocardial ischaemia	5 (5.5)	7 (10.8)		
Lower limb arteriopathy	3 (3.3)	4 (6.1)		
Stroke/transient ischemic attack	2 (2.2)	4 (6.1)		
Carotid stenosis	1 (1.1)	5 (7.7)		
Heart failure	0	4 (6.1)		
Atrial fibrillation	0	3 (4.6)		
Tumor localization, *n* (%)				
Oral cavity	67 (73.6)	57 (87.7)	0.05	0.039
Oropharynx	18 (19.7)	7 (10.7)		
Parotid	2 (2.2)	1 (1.5)		
Sinus	1 (1,1)	0		
Hypopharynx	1 (1.1)	0		
Larynx	1 (1.1)	0		
Nose	1 (1.1)	0		
Preliminary oncologic				
treatment	22%	33%	0.17	0.272
Preoperative morphine				
consumption	17 (18.9)	10 (15.4)	0.57	0.184

No difference in diabetes, cardiovascular, and pulmonary history (distribution), preoperative pain and opioid consumption, preoperative albumin, and radionecrosis was noticed; SMD = standard mean difference; SAS = sleep apnea syndrome. SMD value in bold is statistically significant, n = number. COPD = chronic obstructive pulmonary disease.

**Table 3 medicina-59-00722-t003:** Intraoperative events.

	Group 1 (*n* = 91) Routine	Group 2 (*n* = 65) Optimized	*p* Value	SMD
Duration of surgery (min ± SD)	557 ± 101	607 ± 140	0.01	0.154
Morphine mean mg/kg ± SD	0.16 ± 0.06	0.11 ± 0.04	<0.0001	0.184
Intraoperative fluid				
Volume administered mL/kg/h mean ± SD	10.2 ± 3.1	9.1 ± 2.7	0.02	0.283
Total volume administered mean ± SD	8.2 ± 2.2	8.4 ± 2.4	0.34	0.315
Intraoperative transfusion, yes				
*n* (%)	61 (67.8)	34 (53.1)	0.09	0.303
Ventilatory parameters				
Mean FiO_2_ (%) ± SD	41 ± 4	34 ± 6	<0.0001	0.6
Protective ventilation, yes, *n* (%)	89 (97)	62 (95)	0.03	0.35
Mean PEEP (mmHg)	6 ± 1	6 ± 1		0

FiO_2_-inspired oxygen fraction, PEEP: positive end expiratory pressure, and SMD values in bald are statistically significant.

**Table 4 medicina-59-00722-t004:** Postoperative pulmonary complications.

	Group 1 (*n* = 91) Routine	Group 2 (*n* = 65) Optimized	*p* Value
Respiratory complications, yes, *n* %			0.03
Hypoxemic atelectasis	5	2	
Bronchial superinfection	6	5	
Pulmonary infection	18	6	
Pleural effusion	1	0	
Pulmonary congestion and isolated hypoxemia	4	1	
Respiratory complications by Clavien–Dindo grade			
Grade I	7	1	
Grade II	24	10	
Grade III	0	0	
Grade IV	3	3	

**Table 5 medicina-59-00722-t005:** Multivariate analysis.

	Model Construction	Number of Adjusting Variables	AIC	BIC	Number of Patients	Effect of PAV (OR and IC95)
Model A	Adjustment on all variables having SMD > 0.1	13	177.65	220.08	156	0.37 (0.15–0.94)
Model B	Descending stepwise regression of model A	5	166.24	184.43	156	0.38 (0.16–0.92)
Model C	Adjustment on all variables having SMD > 0.15	10	174.34	210.70	156	0.39 (0.16–0.97)
Model D	Descending stepwise regression of model C	5	166.93	185.11	156	0.40 (0.16–0.96)
Model E	Adjustment on all variables having SMD > 0.2	8	171.54	198.82	156	0.40 (0.16–0.98)
Model F	Descending stepwise regression of model E	4	167.01	182.16	156	0.41 (0.17–0.97)

Summary of different model in multivariate analysis, AIC = Akaike information criterion, BIC = Bayesian information criterion. OR = odd ratio.

**Table 6 medicina-59-00722-t006:** Surgical complications.

	Group 1 (*n* = 91) Routine	Group 2 (*n* = 65) Optimized	*p* Value
Surgical complications, yes, *n* (%)	47 (51.6)	42 (64.6)	0.106
Re intervention, *n* (%)			
Yes, *n* (%)	36 (39.6)	32 (49.2)	
Type of surgical complications			
Complete necrosis, *n* (%)	9 (9.9)	7 (10.7)	
Partial necrosis, *n* (%)	16 (17.6)	6 (9.2)	
Hematoma, *n* (%)	11 (12.1)	10 (15.3)	
Sepsis, *n* (%)	10 (12)	15 (23)	
Fstula/leakage, *n* (%)	11 (12.1)	10 (15.3)	

## Data Availability

Available on demand.

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
