# Peer review of "Impact of Preemptive Postoperative Pressure Support Ventilation and Physiotherapy on Postoperative Pulmonary Complications after Major Cervicofacial Cancer Surgery: A before and after Study"

_medicina, 2023, doi:10.3390/medicina59040722_

Round 1

Reviewer 1 Report

Minor suggestions to improve the overall quality:

Despite reports suggesting that pressure support ventilation facilitates weaning from mechanical ventilation in the intensive care unit, few studies have assessed its effects on recovery from anesthesia. The pressure support ventilation during emergence from anesthesia reduces postoperative atelectasis in patients undergoing laparoscopic surgery using the Trendelenburg positive. The incidence of postoperative atelectasis was lower in patients undergoing either laparoscopic colectomy or robot-assisted prostatectomy who received pressure support ventilation during emergence from general anesthesia compared to those receiving intermittent manual assistance. please discuss and cite doi: 10.1097/ALN.0000000000003997.

CPAP can significantly improve postoperative dyspnea in lung cancer patients. please discuss and cite PMCID: PMC8129302.

- Cancer surgery could need more supplementation and ventilation depending on the reconstruction, the involvement of the upper or lower respiratory airway. please discuss and cite PMID: 31960661 and doi: 10.2217/fon-2019-0053.

Author Response

-- 

We thank the reviewer for his suggestions permitting us to improve the manuscript , we provided new discussions and inserted all references suggested (red in the main text)

Despite reports suggesting that pressure support ventilation facilitates weaning from mechanical ventilation in the intensive care unit, few studies have assessed its effects on recovery from anesthesia. The pressure support ventilation during emergence from anesthesia reduces postoperative atelectasis in patients undergoing laparoscopic surgery using the Trendelenburg positive. The incidence of postoperative atelectasis was lower in patients undergoing either laparoscopic colectomy or robot-assisted prostatectomy who received pressure support ventilation during emergence from general anesthesia compared to those receiving intermittent manual assistance. please discuss and cite doi: 10.1097/ALN.0000000000003997.

Despite reports suggesting that pressure support ventilation facilitates weaning from mechanical ventilation in the intensive care unit, few studies have assessed its effects on recovery from anesthesia. In a randomized study by Jeong et al (3) focused on emergence from anesthesia showed when compared to spontaneous emergence, the pressure support ventilation reduced postoperative atelectasis in patients undergoing laparoscopic colectomy or robot-assisted prostatectomy surgery. 

- CPAP can significantly improve postoperative dyspnea in lung cancer patients. please discuss and cite PMCID: PMC8129302.

- Cancer surgery could need more supplementation and ventilation depending on the reconstruction, the involvement of the upper or lower respiratory airway. please discuss and cite PMID: 31960661 and doi: 10.2217/fon-2019-0053.

The use of continuous airway positive pressure (CPAP) which is in fact is less invasive procedure could also be discussed in this context(13) however  the higher risk  of  barotraumatism especially in this category of patients with  high percentage of COPD history should also be balanced. Accordingly, due the positive results of this quality assurance program the study inclusion protocol is now the protocol for our PACU/SICU admission. Nevertheless we are aware that some complex surgeries with lesser duration or  cervicofacial interventions in relation  to other  types of cancers may need other types of assistance (14) and our protocol is mainly adapted to our specific category of patients needing  mostly free flap reconstruction. Indeed the involvement of lower or upper respiratory tract can totally change reconstruction modalities (15) therefore transitory tracheostomy may not be necessary, allowing facemask noninvasive ventilatory assistance or even CPAP in selected patients.

Reviewer 2 Report

The manuscript suggest that preemptive postoperative pressure support ventilation, physiotherapy and critical respiratory support decrease the incidence of postoperative pulmonary complications in cervicofacial cancer surgery.

1:Punctuate error. Line 238, 311, 312 and 321.

2:Change reference of 8.

Author Response

1:Punctuate error. Line 238, 311, 312 and 321.

2:Change reference of 8.

Thank you for your suggestions  permitting us to imrove the manuscript  , we verified  and corrected  error  punctuations  suggested and removed reference 8 and the sentence related  as requested.

Reviewer 3 Report

The study itself is well executed, the text is easy to understand, and the study protocols can be followed. The field interests anesthesiologists, intensive care doctors, and maxillofacial, reconstructive plastic, and ENT surgeons.

1.       Do the authors routinely admit patients with published characteristics (4-hour surgery and tracheotomy) into ICU (or, as stated, PACU and SICU) for postop care? To rephrase the question, is the study inclusion protocol also the protocol for PACU/SICU admission?

2.       The author's report of a standard blood test is performed daily. Please elaborate on a general test (WBC, electrolytes, etc.).

3.       The authors provide information about blood gas tests performed during the surgery and in the ICU. Please elaborate on whether the patients had an arterial line during the first phase.

4.       In Table 2, the value without »P=« might be sufficient, as well as the delineation of statistically significant values denoted by bold or italics.

5.       In Table 2, strict use of the same type of quotation of values as proposed in Question/Remark 4.

6.       In Table 2, the authors probably binarized Pulmonary disease and Cardiovascular disease to attain detailed data for statistical analysis. Maybe a cumulative number (of binarization) might be in order. As for Pulmonary disease, yes, number in Group 1  and a number in Group 2. If the authors used any other type of grouping, a short elaboration for the readers should be included at the end of Table 2.

7.       The same proposal of strict use of similar/exact delineation of results goes for Tables 3-5.

8.       The authors in Results refer to multivariate analysis performed using 6 statistical models they have included in Appendix. The modeling is informative, and the results seem valid. Therefore their inclusion in the main body of results may be quite welcome for a reader.

9.        In Lines 330-331, the authors state a 10-hour surgery cut-off to expect increased risk for pulmonary complication. Would it be possible to show the cut-off line in Figure 2?

10.   In Line 366, Group 2 instead of Group2 is advised.

11.   The exclusion criteria in studies involving cancer patients usually exclude children. In Line 380, the authors state that 2 children were excluded. Most parameters would significantly change; therefore, this statement would more adequately belong to Figure 2: The study flowchart.

Author Response

The study itself is well executed, the text is easy to understand, and the study protocols can be followed. The field interests anesthesiologists, intensive care doctors, and maxillofacial, reconstructive plastic, and ENT surgeons.

Thank you so much for this critical review permitting us to improve the manuscript , we tried to respond to all your concerns and suggestions 

  1. Do the authors routinely admit patients with published characteristics (4-hour surgery and tracheotomy) into ICU (or, as stated, PACU and SICU) for postop care? To rephrase the question, is the study inclusion protocol also the protocol for PACU/SICU admission?                                                 Thank you for your suggestion , yes the study inclusion  protocol is the protocol for PACU /SICU admission , we added this sentence in the discussion section  

    All patients stayed in post anesthetic care unit (PACU) followed by surgical care unit  (SICU) according to our routine local clinical protocol.

      In addition , we added a sentence  in the discussion as the ventilatory protocol is now part of our routine clinical  protocol

  1. The author's report of a standard blood test is performed daily. Please elaborate on a general test (WBC, electrolytes, etc.).                                 Thank you for this suggestion we added more details as asked               

    . If respiratory events were suspected a chest x ray was performed in conjunction with blood tests including C reactive protein, electrolytes,  white blood cells, platelets counts , hemoglobin level and glomerular filtration rate were performed daily during the stay in surgical intensive care unit (SICU), blood gas was performed if requested depending on clinical situations.

  2. The authors provide information about blood gas tests performed during the surgery and in the ICU. Please elaborate on whether the patients had an arterial line during the first phase.                                                          The arterial line was inserted in most cases , however the insertion of arterial  line was not systematic and not related to the protocol , but to the type of surgery and patients comorbidites. we added as requested a special paragraph in the method section                                                

     Intraoperative monitoring consisted of electrocardiogram,  non-invasive blood pressure monitoring, oxygen saturation,  objective neuromuscular monitoring and bispectral index and intraoperative temperature  while respiratory parameters was recorded from the anesthesia machine including ventilatory  pressures  , ventilatory volume and  end tidal PCO2  for all patients. Invasive arterial line if necessary according to our local protocol for free flap surgery ASA III patients,  long or complex surgeries, and patients with major cardiorespiratory and renal history.   

  3. In Table 2, the value without »P=« might be sufficient, as well as the delineation of statistically significant values denoted by bold or italics.          Thank you for your suggestion , we removed most of  the p values and the  SMD result are presented in bald if significant as requested.
  4. In Table 2, strict use of the same type of quotation of values as proposed in Question/Remark 4. done 
  5. In Table 2, the authors probably binarized Pulmonary disease and Cardiovascular disease to attain detailed data for statistical analysis. Maybe a cumulative number (of binarization) might be in order. As for Pulmonary disease, yes, number in Group 1  and a number in Group 2. If the authors used any other type of grouping, a short elaboration for the readers should be included at the end of Table 2.      yes  we compared distribution in percentage  of disease in each group , this is now shortly mentionned in table 2                                         
  6. The same proposal of strict use of similar/exact delineation of results goes for Tables 3-5.                                Thank you for your suggestion we modified accordingly all the mentionned tables  to be consistent in the presentation. 
  7. The authors in Results refer to multivariate analysis performed using 6 statistical models they have included in Appendix. The modeling is informative, and the results seem valid. Therefore their inclusion in the main body of results may be quite welcome for a reader.                               You are right , we deleted the appendix and inserted  the table in the main text thank you for this suggestion  
  8.  In Lines 330-331, the authors state a 10-hour surgery cut-off to expect increased risk for pulmonary complication. Would it be possible to show the cut-off line in Figure 2?                                                                      Thank you we added a cut off line and a phrase in the legend  of the figure to explain it 
  9. In Line 366, Group 2 instead of Group2 is advised.
  10. The exclusion criteria in studies involving cancer patients usually exclude children. In Line 380, the authors state that 2 children were excluded. Most parameters would significantly change; therefore, this statement would more adequately belong to Figure 2: The study flowchart.                             Absolutely, we added a statement for the flowchart and deleted it from the discussion